# Genome-Wide Identification and Expression Analysis of UBiA Family Genes Associated with Abiotic Stress in Sunflowers (*Helianthus annuus* L.)

**DOI:** 10.3390/ijms24031883

**Published:** 2023-01-18

**Authors:** Mingzhe Sun, Maohong Cai, Qinzong Zeng, Yuliang Han, Siqi Zhang, Yingwei Wang, Qinyu Xie, Youheng Chen, Youling Zeng, Tao Chen

**Affiliations:** 1Xinjiang Key Laboratory of Biological Resources and Genetic Engineering, College of Life Science and Technology, Xinjiang University, Urumqi 830017, China; 2Zhejiang Provincial Key Laboratory for Genetic Improvement and Quality Control of Medicinal Plants, College of Life and Environmental Science, Hangzhou Normal University, Hangzhou 311121, China

**Keywords:** *UBiA*, genome-wide analysis, abiotic stress, *Helianthus annuus* L.

## Abstract

The *UBiA* genes encode a large class of isopentenyltransferases, which are involved in the synthesis of secondary metabolites such as chlorophyll and vitamin E. They performed important functions in the whole plant’s growth and development. Current studies on UBiA genes were not comprehensive enough, especially for sunflower *UBiA* genes. In this study, 10 *HaUBiAs* were identified by domain analysis these *HaUBiAs* had five major conserved domains and were unevenly distributed on six chromosomes. By constructing phylogenetic trees, 119 *UBiA* genes were found in 12 species with different evolutionary levels and divided into five major groups, which contained seven conserved motifs and eight UBiA subsuper family domains. Tissue expression analysis showed that *HaUBiAs* were highly expressed in the roots, leaves, and seeds. By using promoter analysis, the cis-elements of *UBiA* genes were mainly in hormone signaling and stress responses. The qRT-PCR results showed that HaUBiA1 and HaUBiA5 responded strongly to abiotic stresses. Under ABA and MeJA treatments, HaUBiA1 significantly upregulated, while HaUBiA5 significantly decreased. Under cold stress, the expression of UBiA1 was significantly upregulated in the roots and stems, while UBiA5 expression was increased only in the leaves. Under anaerobic induction, UBiA1 and UBiA5 were both upregulated in the roots, stems and leaves. In summary, this study systematically classified the UBiA family and identified two abiotic stress candidate genes in the sunflower. It expands the understanding of the UBiA family and provides a theoretical basis for future abiotic stress studies in sunflowers.

## 1. Introduction

Many biologically active substances must be prenylated to become soluble in the membrane to become functional [1,2,3]. UBiA superfamily is a class of proteins that fuse isoprenoid or phytyl to aromatic compound molecules from the aqueous phase of the cell. Proteins with UBiA structural domains are widely found in bacteria, fungi, and plants [4]. In plants, these PTases (prenyltransferases) are integral membrane proteins that contain at least one aspartate-rich motif (e.g., NDxxDxxxD) for the binding of prenyl diphosphate via Mg^2+^ ions or similar cations, such as Mn^2+^, Co^2+^ or Ni^2+^ [5]. The substrates of the prenylation reaction play essential roles in all species. The prenylation reaction is generally an evolutionarily conserved and rate-limiting step of the secondary metabolite biosynthetic pathway. The compounds that can be prenylated by PTase include 4-hydroxybenzoic acid (PHB), homogeneous acid (HGA), 1,4-dihydroxy-2-naphthalene acid (DHNA), flavonoids chlorophyll or proheme, which play important roles in electron transportation, anti-oxidation, and formation of structural lipids for microbial cell walls and membranes.

PHB Polyprenyltransferase (PPT) has been the most comprehensively studied isopentenyl transferase, which transfers varying lengths of isopentenes to PHB [3]. PPT is the rate-limiting step of UQ (*Ubiquinone*) biosynthesis [6]. In *Arabidopsis thaliana*, embryonic retardation is a prominent trait of the UQ-deficient mutant caused by PPT absent in the early stage of zygotic embryogenesis [7]. OsPPT1 exhibits PPT enzymatic activity, enabling the restoration of respiratory ability and UQ synthesis, respectively [8]. The tobacco UQ overexpressed plants showed increased tolerance to oxidative stress induced by methyl violet and salt stress [9].

Based on different isopentenyl donors, homogentisate polyprenytransferase can be divided into three categories, such as homogentisate phytyltransferases (HPT), homogentisate geranylgeranyl transferase (HGGT) and homogentisate solanesyl transferase (HST). HST is involved in the biosynthesis of plastoquinone-9. When α-tocopherol is consumed, plastoquinone-9 can replace α-tocopherol, providing protection against singlet oxygen toxicity [10,11]. HPT and HGGT are key rate-limiting enzymes of Vitamin E synthesis that catalyze the binding of homogentisate acid to different isopentenyl donors. HGGT mostly exists in monocotyledons, and its product has a side chain with three unsaturated bonds [12]. HGGT mutants have wilting phenotypes, and a significant reduction in seed total starch content [13] and rice HPT mutants are sensitive to cold stress [14]. The overexpression of homogentisate phytyltransferase genes in tobacco enhances drought tolerance [15]. Pentenyl flavonoids and isoflavones are mainly accumulated in legumes, which can resist the invasion of pathogens by plants and play an important role in the biosynthesis of the secondary metabolites [16]. ChlorophyII synthase is the final step in chlorophyII biosynthesis, which is the phosphorylation of chlorophyII lactone with phytyl or geranyl [17]. Protoheme IX farnesyltransferase (heme O synthase), connecting the tail of farnesyl to protoheme IX, finally turns heme O to heme A in participating plant respiration. DHNA-phytyltransferases, another member of the UBiA superfamily, are involved in phylloquinone biosynthesis by using DHNA as substrate [18].

Sunflower is the fourth largest oilseed crop in the world and has a high economic value. It is also a pioneer plant in saline areas with strong tolerance to various stresses [19,20]. UBiA transferase were involved in the accumulation of secondary metabolism and were the key rate-limiting enzyme for many important reactions, such as the synthesis of vitamin E chlorophyll and proheme. It plays an important role in plant growth and development. Studying how a sunflower senses and responds to various stresses from nature, can provide new insights and the mechanisms of plant tolerance to stress. However, few studies have been done on the functional analysis of the UBiA family under abiotic stresses. Therefore, this study systematically analyzed the UBiA family in sunflower and other 11 species on different evolutionary nodes. Using a more macroscopic perspective for in-depth analysis of the UBiA genes.

In this study, phylogenetic relationship, conserved domain analysis and other bioinformatics methods were carried out in sunflowers. The *UBiA* genes of species were highly conserved and were divided into five subgroups. The motifs of the encoded proteins performing the same function were clustered in a similar branch. Promoter analysis revealed that the *UBiA* genes in all species contained at least one ABA, MeJA, cold stress and anaerobic stress-induced response element. In the sunflower, the *UBiA* genes were expressed in different tissues. *HaUBiAs* were responsive to ABA, MeJA, cold stress and anaerobic induction. *UBiA1* and *UBiA5* were identified as highly responsive genes to both hormones and stress treatments. Our study revealed the important role of the UBiA family in sunflowers and screened out two candidate genes responding to abiotic stress.

## 2. Results

### 2.1. Identification and Chromosomal Location of UBiA Genes and in Helianthus annuus

To identify the UBiA family members, genome-wide and full-length transcriptome analyses were performed using HMM software. After removing duplicate transcripts and pseudogenes, 10 *UBiA* genes were finally obtained in *Helianthus annuus*. To understand the basic characteristics of *HaUBiA* genes, the gene structures were analyzed (Figure 1A). The phylogenetic relationship and conserved domains of these 10 HaUBiAs were analyzed, and the results showed that the sunflower UBiA had six structural domains, including UBiA-HPT, UBiA-HST, UBiA-COX10, UBiA-PPT, UBiA-ATG4, UBiA-ABC4. HPT had the highest number of four sequences (Figure 1B). Ten *HaUBiA* genes were unevenly distributed on six chromosomes (Figure 1C). Collinearity analyses of burdock lettuce and sunflower were performed (Appendix A). There was a linear relationship in seven pairs of genes between the lettuce and sunflower and four pairs between the burdock and sunflower. The effective length of the coding region sequence is 921~1308 bp. The linearly related gene pairs had a ka/ks ratio between 0.15 to 0.3; it suggested that those genes tend to be selected for purification (Appendix A). This result indicated that the UBiA genes of Asteraceae were highly conserved in evolutionary terms.

Via a sequences blast, a total of 119 *UBiA* genes were extracted and screened from these 12 selected species. Among them, *Glycine max* had a maximum of 24 *UBiA* genes, and *Synechocystis* had a minimum of five *UBiA* genes (Table 1). The basic information of these genes was analyzed and summarized (Table 2), including amino acid number, molecular weight, theoretical PI, instability index, aliphatic index, and the grand average of hydropathicity. Among them, *PpUBiA6* encoded the largest protein, containing 596 amino acids, while *OsUBiA9* encodes the smallest protein, consisting of 205 amino acids. The molecular weight (MW) of the protein was between 21.9–64.2 kDa. The range of isoelectric point (PI) was 6.05–10.63. The range of the instability index was 19.93–58.59. The aliphatic index was 86.33–134.87. The grand average of hydropathicity was −0.025–0.897. Hydrophilic results showed that UBiA usually had hydrophilic properties.

### 2.2. Identification and Phylogenetic Analysis of UBiA Families in 12 Species

To investigate the genetic phylogenetic relationship of the *UBiA* gene family, 12 different species representing evolutionary nodes were selected. According to the evolutionary level from low to high, a temporal tree of these species was constructed. Figure 2A shows the origin time, evolution degree and genetic relationship between 12 different species.

Through phylogenetic tree analysis, the number of *UBiA* members in the first four groups was similar, between 14–16, while the number in Group 5 was as high as 56. This suggested that these *UBiA* members were conserved during the plant evolution, but still, some differences between species had different numbers in each branch. The number of *UBiA* members in Group 5 increased in more evolved species and in dicotyledonous plants; the *UBiA* members expanded to different degrees in this group, especially in soybeans (Figure 2).

### 2.3. Structure Analysis of the UBiA Family Members

To explore the potential functions of the UBiA family members, the conserved motifs and protein domains of 119 UBiA proteins were analyzed (Figure 3). The members of the UBiA family can be divided into five major groups. The classification of *UBiA* family genes is shown in Table 1. The members of each class of the *UBiA* family are present in each species. Group 1 corresponds to *PHB polyprenyltransferase* (PPT), Group 2 corresponds to *protoheme IX farnesyltransferases* (COX10), Group 3 corresponds to *chlorophyll synthases* (ATG4), Group 4 corresponds to *DHNA-phytyltransferases* (ABC4) in phylloquinone biosynthesis, Group 5 corresponds to *Homogentisate polyprenytransferase* (HPT) (Appendix A). To further understand the structural features of *UBiA* family genes, an in-depth analysis of their conserved motifs and structural domains was performed.

For motif analysis, seven different motifs were predicted based on *UBiA* protein sequences (Figure 3B and Appendix A). The length of the motifs varied from 21 to 50 bp. Conserved motifs of genes that perform the same function are arranged in a fixed order. In the case of *PHB polyprenyltransferase*, the two motifs are motif 6 and motif 2. Chlorophyll synthase contains three motifs in the order of: motif 1, motif 7 and motif 2. *DHNA-phytyltransferases* when only motif 2 was present, suggesting that motif 2 is the key motif for *DHNA-phytyltransferases*. The *Protoheme IX farnesyltransferase* contains only motif 6. This indicates that motif 6 is the core motif of this class of enzymes. *Homogentisate polyprenytransferase* consisted of six motifs arranged in the order of motif number 5-1-3-7-2-4.

For domain analysis, all selected genes contained the *UBiA* domain (Figure 3C). In Groups 1 to 4, each group corresponds to one conserved domain. Group 1 corresponds to *PPT*, Group 2 corresponds to *COX10*, Group 3 corresponds to *ATG4*, and Group 4 corresponds to *ABC4*. Group 5 contains four domains, including *HPT, HST, PT, HGGT*. Each species had one or more *UBiA* genes in each of the five groups, except for *HGGT*, which occurred only in *Oryza sativa*. *ATG4* was missing in Group 2 of *Arctium lappa*. These results indicate that the UBiA genes were also highly conserved across species, with clear distinctions between the different structural domains.

### 2.4. Expression Profile of UBiA Genes in Helianthus annuus

To analyze the biological function of 10 *UBiA* genes in *Helianthus annuus*, we performed RT-PCR analysis (Figure 4A). The *UBiA* genes were expressed in the highest amount in the leaves, followed by roots and stems, which were hardly expressed in the floral organs. We wanted to understand the pattern of the *UBiA* genes during seed maturation. Therefore, we selected six time points during the ripening process to analyze the expression of *UBiA* genes in seeds (Appendix A). We found three different expression patterns in seeds. In the first few days after flowering, the expression level of *HaUBiA1* and *HaUBiA2* was high and decreased during the subsequent seed ripening process. The expression of *HaUBiA* genes mostly increased as the seeds matured in an upward trend. The high expression level at the beginning of the seed setting may be related to pollination. The second expression pattern was that these *HaUBiAs*, including *HaUBiA3*, *HaUBiA4*, *HaUBiA7* and *HaUBiA9,* increased gradually when seed maturing was at a maximum at late maturity. The third expression pattern was the genes, such as *UBiA5* and *UBiA6* were upregulated only at one period. In short, we can infer that *HaUBiAs* may play an important role in the process from pollination to seed maturity (Figure 4B).

### 2.5. Cis-Regulatory Elements Analysis of HaUBiA Promoters

To further explore the putative functions of *UBiA* family genes in *Helianthus annuus*, 2000 bp upstream sequences of 119 genes were extracted and analyzed potential cis-elements using PlantCARE (http://bioinformatics.psb.ugent.be/webtools/plantcare/html, accessed on 1 November 2022). The predicted transcription factor (TF) binding patterns were classified into three categories: hormone-related elements, stress-related elements and growth and development-related elements.

We first analyzed the promoter regions of *UBiA* genes in sunflowers (Figure 5A). They all had responsive elements, including MeJA, ABA, cold stress, and anaerobic induction. The same results were seen in the promoter regions of 119 UBiA genes (Figure 5B). MeJA responsiveness contained two different motifs named TGACG-motif and CGTCA-motif, respectively. The ABA response contained 10 different ABRE motifs and the cold stress responsive element had only one LTR motif. The anaerobic stress response contains only one unique ARE motif. These results suggest that *UBiA* is widely involved in plant growth and development and plays an important role in plant response to stress and hormone response and is highly conserved across species.

### 2.6. Expression Patterns of HaUBiA Genes under the Treatments of Different Hormones

Many responsive elements on the upstream *HaUBiA* genes were associated with MeJA and ABA by promoter analysis software, and then qRT-PCR analysis was used to verify whether these genes were responding to the above hormone signals (Figure 6). Except for *HaUBiA8*, the other nine genes showed three different expression patterns after ABA treatment. The first category of genes (seven genes), whose expression increased at the beginning of the treatment and decreased dramatically in the end. The second category of genes (two genes) showed less change during the whole treatment. The third category presented increased expression with the increase in treatment time.

Three expression patterns of *HaUBiA* existed under MeJA treatment. The first category, whose expression increased at the beginning and gradually decreased with treatment time, included five genes. The second category of genes, in which the expression was like the first category but eventually remained higher than before treatment, included four genes. The expression of genes in the third category decreased with increasing treatment time, including only one gene. These results above showed that most *HaUBiA* responded to ABA and MeJA treatment. The results indicate that *HaUBiA* genes were able to respond to ABA and MeJA signaling. The different response patterns suggest that ABA and MeJA may involve in the synthesis of secondary sunflower metabolites and plays a regulatory role to some extent. All these results correspond to the results of promoter analysis and indicate that *HaUBiA* plays an important role in sunflowers.

### 2.7. Expression Analysis of HaUBiA under Different Stresses

The expression pattern of the HaUBiA gene in cold stress and anaerobic induction. Three tissue sites were chosen (roots, stems and leaves) to analyze the expression under stress treatment. During the cold stress treatment, all genes, except for *HaUBiA4* and *HaUBiA9*, were expressed in different tissue, and the expression was significantly changed in at least one tissue during the cold stress treatment (Figure 7A). Among these genes, *UBiA2* was expressed in various tissues, and it was significantly downregulated under cold stress. *UBiA2* may be an important gene widely involved in sunflower cryoprotection. So, to speak, the *HaUBiA* genes had a strong response to the cold stress signal. Under anaerobic induction, the expression of all the *HaUBiA* genes was changed at least in one tissue (Figure 7B). In contrast, *HaUBiA1* and *HaUBiA5* were significantly upregulated in all tissue, and it was speculated that they may serve as important regulatory genes for maintaining organism survival under anaerobic conditions. The results suggested that *HaUBiA* genes were strongly induced under cold stress and anaerobic induction. It maybe plays an important function in sunflower resistance to adversity stress.

## 3. Discussion

*UBiA* family genes are widely involved in the synthesis of various secondary metabolites in plants. However, the evolution of *UBiA* family genes in the plant is not clear. In this study, 119 *UBiA* genes from 12 species with different degrees of evolution were selected to analyze phylogenetic relationships, structural domain composition, gene structure and cis-acting elements. In addition, the responses and the potential functions of *HaUBiA* family genes to cold stress, anaerobic induction, MeJA, and ABA were detected and explored. By constructing phylogenetic trees, a clear classification among UBiAs with five major clades with 119 UBiA family members of 12 species. Each branch corresponds to different functions, and these members were distributed among species at different levels of evolution. The number of UBiA members increased in plants with higher levels of evolution. In soybean, new functions emerged in UBiA, but the number of branches did not change. This suggested that the UBiA family of members had evolved to some extent during evolution, but natural selection is still relatively conserved. A comprehensive analysis of structural domains and motifs showed that most UBiA members contained seven conserved motifs, the most conserved of which was NDxxDxxxD. Interestingly, the conserved motifs of the same functional members between species were arranged in a certain order. This rule applied to most of the UBiA members. The tissue expression results showed that the *UBiA* genes were expressed in various tissues, with higher expression in the roots, stems and leaves, and accumulated during seed maturation.

In soybean, secondary metabolic genes acquire new functions from primary metabolic genes through positive selection driving gene replication and new functionalization [21]. Therefore, the expansion of the *UBiA* gene in Compositae also has the possibility to drive new functions. Transcriptional regulation plays a dominant role in the regulation of gene expression and is mainly controlled by its cis-acting elements at the gene promoters [22]. To explore the potential functions of *UBiA* family genes, we analyzed the cis-acting elements of all *UBiA* family promoter regions. The results showed that the largest proportion of cis-acting elements responded to MeJA, ABA and anaerobic induction, and low temperature and this result was consistent in all selected species, suggesting that the promoters of the *UBiA* genes are conserved and highly responsive to hormone treatment. Our study detected many ABA- and MeJA-responsive elements in the *Helianthus annuus UBiA* family. We selected all 10 *UBiA* genes in sunflower and tested their expression by RT-qPCR analysis under cold, anaerobic induction MeJA and ABA treatments. In hormone treatment, most genes in all 10 *HaUBiA* genes undergo expression changes in the leaves. Sunflower responds to treatments with both hormones. ABA and MeJA signaling can promote stomatal closure and regulate the expression of related genes [23,24]. Therefore, we hypothesized that sunflower drought resistance might be due to the hormones involved in regulating stomatal closure. In addition, several genes were significantly induced in the leaves, stems, and roots during anaerobic and cold stress, indicating that these genes are involved in hormone response and play important functions in resistance to adversity stress.

Mutations in rice HPT, a key gene for vitamin E synthesis, showed overall growth retardation throughout the growth period, which caused most agronomic traits to be damaged and cold tolerance reduced [14]. In lettuce, *HPT* was preferentially expressed in mature leaves. The expression of *LsHPT* was significantly increased under drought and high light stress [25]. Reactive oxygen species (S), a singlet oxygen that affects lipids, proteins and nucleic acids [26,27]. Using HGA as a substrate, plants can synthesize α-tocopherol (HPT catalyze) and plastochromanol-8 (HST catalyze). Tocopherol can be used as a singlet oxygen scavenger in photosystem II [18]. The unsaturated side chains of plastochromanol-8 also have quenching activity for singlet oxygen in the hydrophobic environment [28]. Chlorophyll synthase (UBiA3) can also indirectly affect the synthesis of vitamin E. In *Arabidopsis*, 80% of PDP (prenoid phytyl diphosphate) synthesis must be catalyzed by Chlorophyll synthase [29]. NAC transcription factors regulate cold stress in *Arabidopsis* [30]. In *Arabidopsis*, the homologs gene of HaUBiA5 were able to bind to NAC transcription factors [31]. HaUBiA5 might bind to NAC proteins (Appendix A). We can speculate that UBiA5 may be regulated by NAC transcription factors in leaves.

*UBiAs* not only play an important role in plants but are also necessary to maintain the growth of animals. COX10 mutations in humans trigger mitochondrial causes such as Leigh syndrome [31,32], a factor that may increase the risk of Alzheimer’s disease [33]. PBH isopentenyl transferase mutation is the cause of multisystem disease and nephropathy in infants [34,35]. In summary, the *UBiA* gene plays an important role in the growth and development of both animals and plants and is an indispensable key gene. In this study, we systematically analyzed the *UBiA* genes among different species using sunflowers as the entry point at the species evolutionary level and provided a molecular basis for an in-depth study of *HaUBiA* genes.

## 4. Materials and Methods

### 4.1. Identification of the UBiA Members in Helianthus annuus *L.*

The genome files and gene annotation files used in this study were from NCBI (https://www.ncbi.nlm.nih.gov, accessed on 1 November 2022) and China National Gene Bank archive system, respectively (https://ftp.cngb.org/pub/CNSA, accessed on 1 November 2022). Information on the conserved structural domain of UBiA (Pfam: PF01040) was obtained from the Pfam Protein Family Database database (http://pfam.xfam.org, accessed on 1 November 2022) [36]. A hidden Markov model was used to evaluate the UBiA domain, and the UBiA family genes of 12 different species were identified with an E value < 10^−5^. CDS was extracted from 12 genomes and translated into proteins. After duplicate transcripts were removed, 119 protein sequences were obtained [37]. The protein parameter CalC of TBtools software (https://github.com/CJ-Chen/TBtools, accessed on 2 November 2022) was used to analyze the number of amino acids, molecular weight, theoretical pI, instability index, aliphatic index, and grand average of hydropathicity.

### 4.2. Phylogenetic Tree Analysis of UBiA Proteins

Protein sequences of UBiA members were compared using the ClustalW program. Phylogenetic trees were constructed by the neighbor-joining method with UBiA members and their homologous sequences in different species using MEGA software (version 7.0). Tree nodes were evaluated through 1000 repeated bootstrap analyses [38]. The final version of phylogenetic tree was showed using ITOL (https://itol.cmbl.del, accessed on 3 November 2022) modification [39].

### 4.3. Chromosomal Location Density, Collinearity, and Ka/Ks Analysis of UBiA Genes

Based on gene annotation information, TBtools was used to analyze the chromosome location and chromosome density of *Helianthus annuus* L. UBiA family genes. The data source is based on the NCBI database. *Helianthus annuus*, *Lactuca sativa* and *Arctium lappa* were selected to analyze the collinearity of the UBiA family among compositae plants using the multicollinearity scan toolkit. Ka and Ks are calculated using the simple Ka/ Ks calculator in TBtools [37,40].

### 4.4. Gene Structure, Conserved Motifs in Promoter and Three-Dimensional Structure

UBiA family structure analysis was analyzed based on gene annotation information and visualized by TBtools software [41]. The domain analysis of this family was analyzed by Batch CD-Search (https://www.ncbi.nlm.nih.gov/Structure/bwrpsb/bwrpsb.cgi, accessed on 5 November 2022) [42]. The online website, MEME (https://meme-suite.org/meme/tools/meme, accessed on 5 November 2022), was used to analyze the conserved motif [43]. The three-dimensional structure of the UBiA family was predicted by UniProt (https://www.uniprot.org/uniprotkb/, accessed on 5 November 2022) [44].

### 4.5. Cis-Regulatory Elements Analysis in UBiA Promoters

The promoter sequences (2000-bp upstream of *UBiA* gene) were obtained and used to predict cis-acting elements by PlantCARE (http://bioinformatics.psb.ugent.be/webtools/plantcare/html/, accessed on 5 November 2022) [45].

### 4.6. Plant Materials and Stress Treatments

In this study, AZB, a variety of *Helianthus annuus*, was used as experimental material to analyze the expression pattern of UBiA family members. Fourteen-day-old seedlings were treated with 15 μmol/L ABA and 200 μmol/L MeJA. Leaves were then collected for RNA extraction and RT-qPCR analysis. For stress treatment, the plant materials were placed in a 4 °C environment and given 16 h of light and 8 h of darkness for 2 days for low-temperature treatment; two-week-old seedlings were completely submerged in water with 16 h of light and 8 h of darkness for one day for anaerobic stress treatment. After treatment, roots, stems, and leaves were collected for RNA extraction and qRT-PCR analysis.

### 4.7. RNA Extraction and Quantitative/Real-Time-PCR (RT-qPCR) Analysis

The total RNA of plant materials was extracted using the OmniPlant RNA Kit to obtain cDNA for subsequent experiments. Use CFX384 real-time system (Biorad, Hercules, CA, USA) to complete qRT-PCR experiment. The reagent used was ChamQ universal SYBR qPCR Master Mix (Vazyme, Nanjing, China). The experiment was repeated three times. The internal reference gene was the tubline of *Helianthus annuus* L. The list of primers used in the experiment is in Appendix A.

## 5. Conclusions

In this study, a systematic analysis of the *UBiA* family was performed. The conservation of *UBiA* genes across species was verified. Two candidate genes of abiotic stresses were screened in the sunflower. In summary, our study has expanded the understanding of *UBiA* genes. It provides a reference for abiotic stress studies in sunflowers and offers new research perspectives for crop breeding. However, the functions of HaUBiA1 and HaUBiA5 in abiotic stress will require further study.

## Figures and Tables

**Figure 1 ijms-24-01883-f001:**
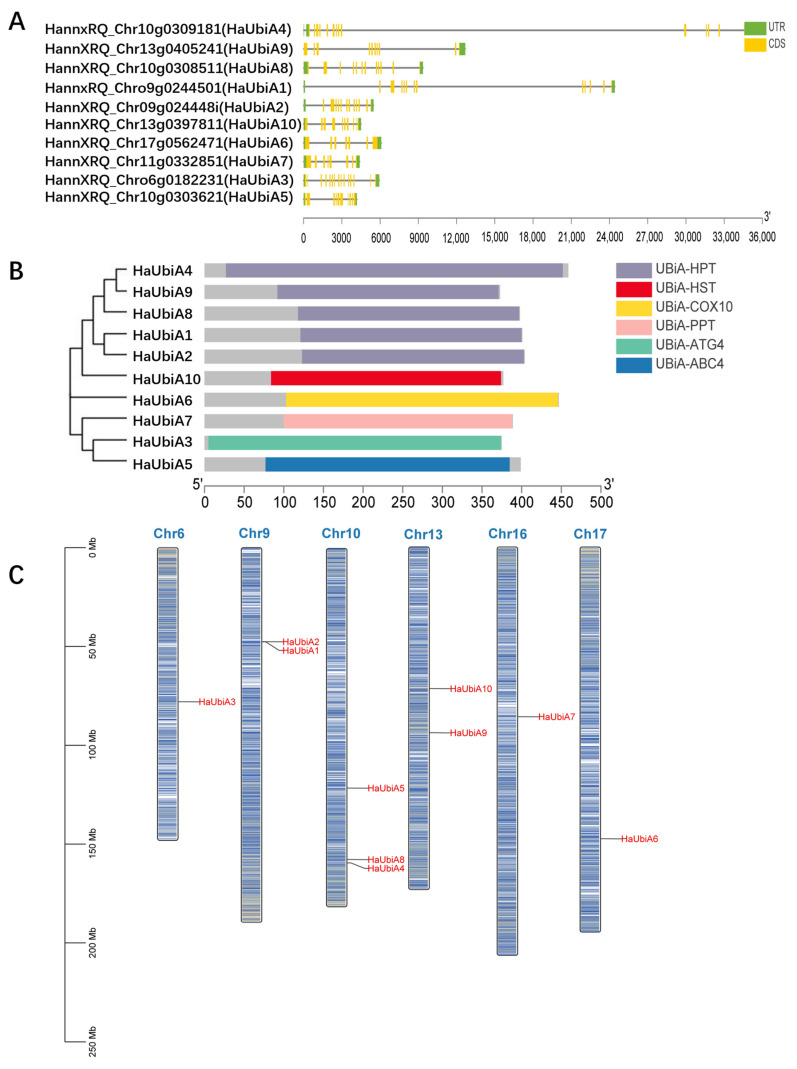
Phylogenetic analysis and chromosomal location of UBiA family genes in *H. annuus*. (**A**) *HaUBiA* gene structures. The green part is the untranslated region, and the yellow part represents the coding sequences. (**B**) Phylogenetic trees of HaUBiAs were constructed by the neighbor-joining method with 1000 bootstrap replications. The grey section indicates the full length of the protein. The *UBiA* genes conserved domain were presented in different colors. (**C**) Chromosomal locations of 10 *HaUBiA* genes. The scale bar on the left indicates the length (Mb) of *Helianthus annuus* chromosomes.

**Figure 2 ijms-24-01883-f002:**
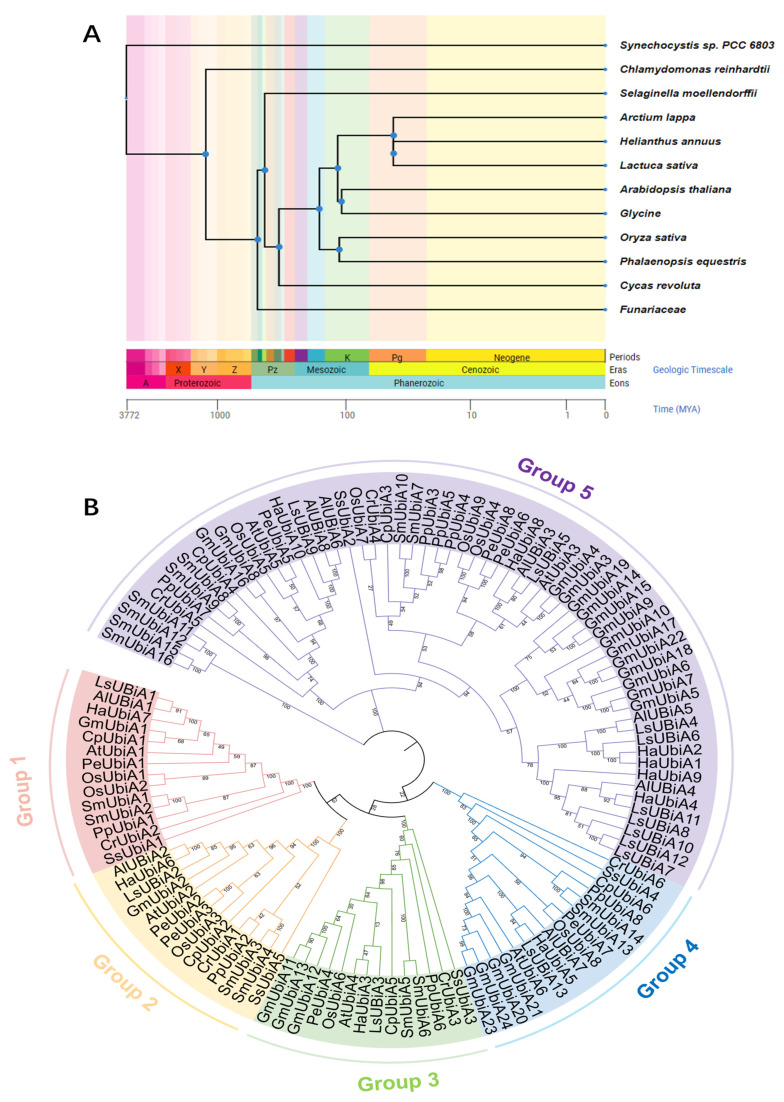
Species differentiation time and UBiA phylogenetic tree of 12 species. (**A**) The species time tree was constructed according to the origin time of 12 species. The colour blocks on the species time tree represent the geological time of their origin. (**B**) Phylogenetic trees of 12 species were constructed using MEGA7 by the neighbor-joining method with 1000 bootstrap replications. Each member cluster was labeled with different colors.

**Figure 3 ijms-24-01883-f003:**
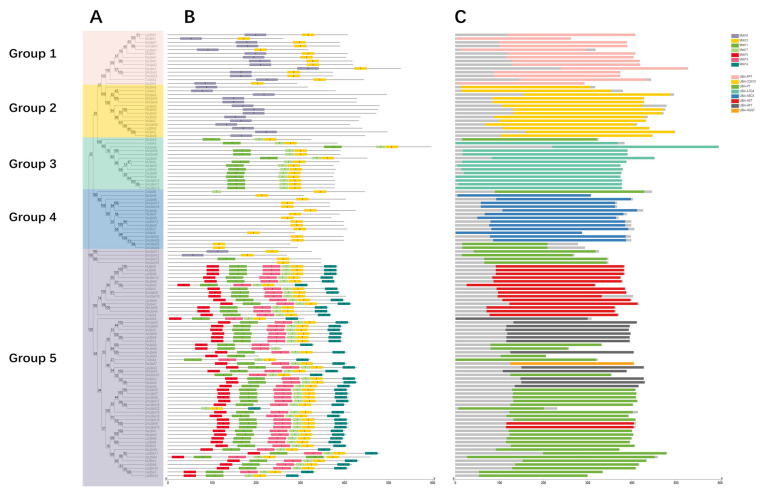
Conserved motifs and protein domain in 119 UBiAs. (**A**) Phylogenetic tree of UBiAs family members. (**B**) Conserved motifs of UBiA proteins. Seven conservative motifs of UBiAs were identified by using the MEME online tool. Each motif in UBiA proteins was represented with different colors. (**C**) Conserved domains of UBiA proteins. A total of eight protein domains were identified using NCBI conserved domain database and displayed in different colors.

**Figure 4 ijms-24-01883-f004:**
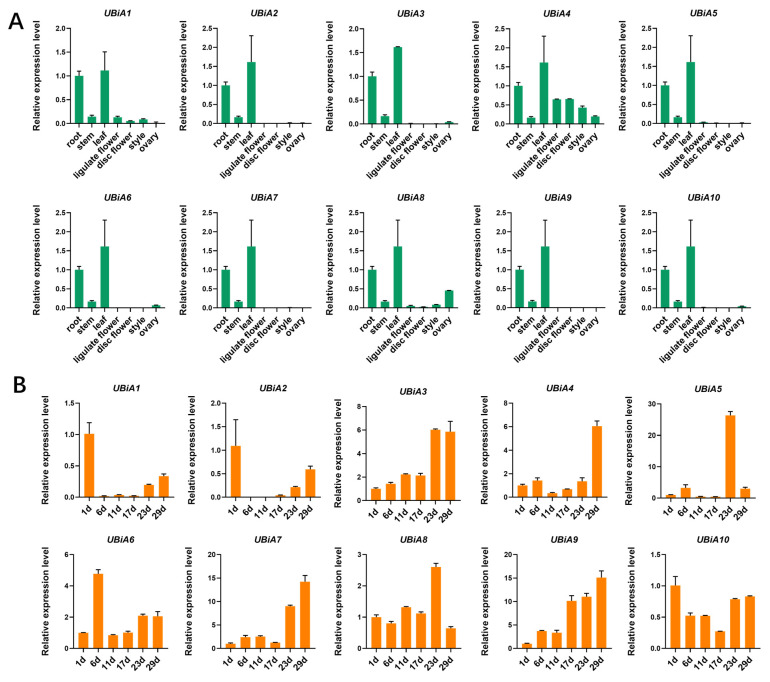
Expression analysis of 10 *HaUBiA* genes in different tissues and at different stages of seed maturity in *H. annuus*. (**A**) The relative expression of 10 *HaUBiA* genes in different tissues using qRT-PCR. (**B**) Relative expression of 10 *HaUBiA* genes in seeds. Six time points were designed from the first day after flowering to the full maturity of the seeds. *HaTubulin* was used as an internal control. Values were means ± SD *(n* = 3). (Student’s *t*-test).

**Figure 5 ijms-24-01883-f005:**
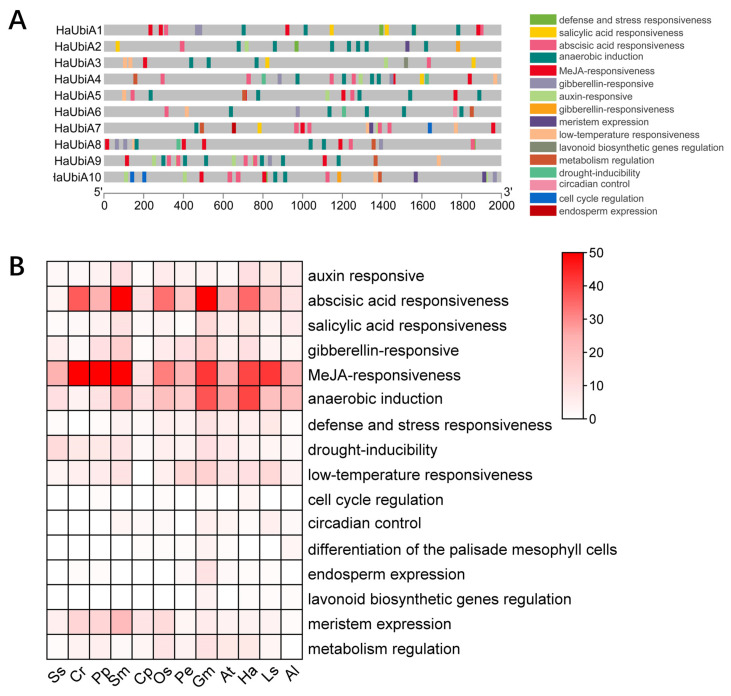
Cis-element analysis of *UBiAs* promoter. (**A**) Cis-acting elements of *UBiA* genes promoters in *H. annuus.* The part in grey were 2000 bp upstream from *UBiAs*. (**B**) Cis elements of *UBiA* promoters in the selected 12 species were collected and classified. The number of each element was indicated by the depth of the red color. Ss—*Synechocystis*; Cr—*Chlamydomonas reinhardtii*; Pp—*Physcomitrium patens*; Sm—*Selaginella moellendorffii*; Cp—*Cycas revoluta*; Os—*Oryza sativa*; Pe—*Phalaenopsis equestris*; Gm—*Glycine max*; At—*Arabidopsis thaliana*; Ha—*Helianthus annuus*; Ls—*Lactuca sativa*; Al—*Arctium lappa*.

**Figure 6 ijms-24-01883-f006:**
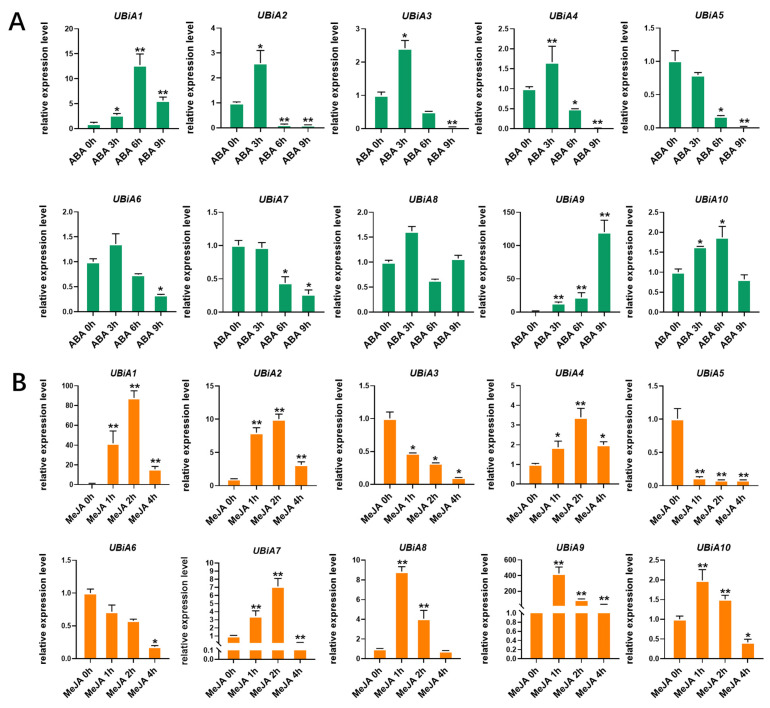
Expression analysis of 10 *UBiA* genes in leaves under ABA and MeJA hormone treatment in *H. annuus*. (**A**) 50 μmol/L ABA treatment. (**B**) 200 μmol/L MeJA treatment. Fourteen-day-old seedlings of sunflowers were used to test the response of hormone signals. *HaTubulin* was used as an internal control. Values were presented as means ± SD *(n* = 3). (* *p* < 0.05, ** *p* < 0.01, Student’s *t*-test).

**Figure 7 ijms-24-01883-f007:**
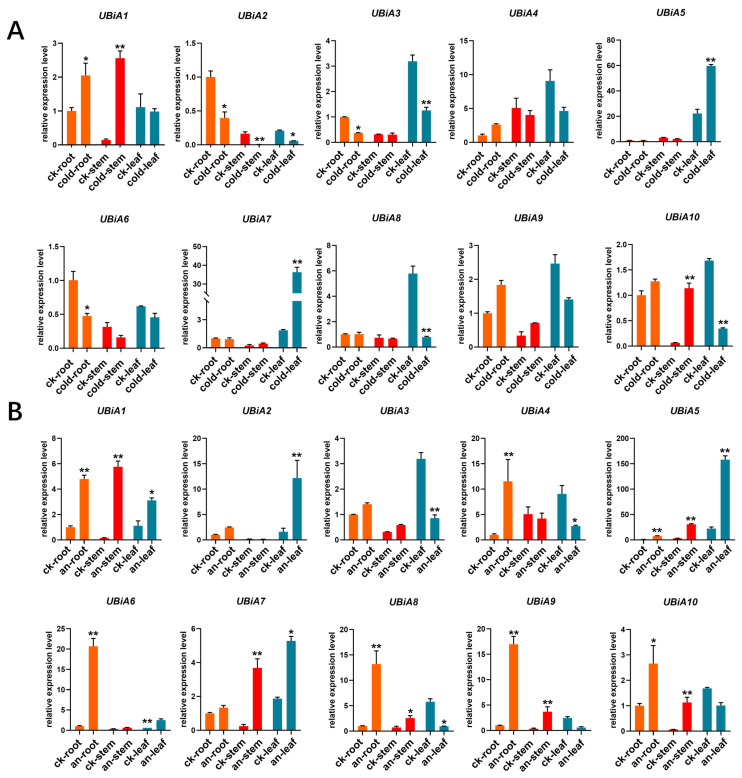
Expression patterns of *HaUBiA* genes in the roots, stems, and leaves under abiotic stress. (**A**) Under the treatment of low temperatures. (**B**) Under the treatment of anaerobic stress. *HaTubulin* was used as an internal control. Values were presented as means ± SD (*n* = 3). (* *p* < 0.05, ** *p* < 0.01, Student’s *t*-test).

**Table 1 ijms-24-01883-t001:** Classification of UBiA family proteins.

Species	I	II	III	IV	V	Total No.
*Synechocystis*	1	1	1	1	1	5
*Chlamydomonas reinhardtii*	1	1	1	1	2	6
*Physcomitrium patens*	1	1	1	1	4	8
*Selaginella moellendorffii*	4	4	2	2	4	16
*Cycas revoluta*	1	1	1	1	2	6
*Oryza sativa*	2	1	1	1	4	9
*Phalaenopsis equestris*	1	1	1	2	3	8
*Glycine max*	1	3	4	1	15	24
*Arabidopsis thaliana*	1	1	1	1	2	6
*Helianthus annuus*	1	1	1	1	6	10
*Lactuca sativa*	1	1	2	1	8	13
*Arctium lappa*	1	0	1	1	5	8
Total No.	16	16	17	14	56	119

**Table 2 ijms-24-01883-t002:** The characteristics of UBiA family proteins.

ID	Rename	Number of Amino Acids	MW (Da)	PI	Instability Index	Aliphatic Index	Grand Average of Hydropathicity	LOCATION
XM_006602661.3	*GmUBiA1*	389	42,818.65	9.51	39.86	97.56	0.192	Plasma Membrane
XM_003556504.5	*GmUBiA2*	411	44,599.15	9.53	46.35	86.33	0.098	Plasma Membrane
NM_001251443.2	*GmUBiA3*	411	46,172.21	9.46	41.75	106.28	0.477	Plasma Membrane
NM_001254567.1	*GmUBiA4*	395	44,364.22	9.37	43.28	109.82	0.532	Plasma Membrane
XM_003541408.5	*GmUBiA5*	408	46,233.75	9.65	46.56	106.59	0.42	Plasma Membrane
XM_006588774.4	*GmUBiA6*	409	46,464.65	9.45	43.87	104.43	0.414	Plasma Membrane
XM_003541409.4	*GmUBiA7*	410	46,311.92	9.66	49.08	109.61	0.438	Plasma Membrane
NM_001250971.1	*GmUBiA8*	389	43,362.87	9.77	38.49	107.89	0.491	Plasma Membrane
NM_001249061.2	*GmUBiA9*	409	46,011.81	9.36	38.15	96.36	0.308	Plasma Membrane
XM_041014270.1	*GmUBiA10*	413	47,079.83	9.56	53.68	112.4	0.443	Plasma Membrane
NM_001252704.2	*GmUBiA11*	377	40,923.53	7.67	29.12	107.67	0.336	Plasma Membrane
XM_003552213.4	*GmUBiA12*	377	40,981.45	6.9	33.9	104.8	0.3	Plasma Membrane
XM_014768972.3	*GmUBiA13*	377	40,959.48	8.31	30.94	105.86	0.308	Plasma Membrane
NM_001366992.1	*GmUBiA14*	408	45,830.1	9.27	51.83	114.24	0.526	Plasma Membrane
XM_041017800.1	*GmUBiA15*	408	45,834.76	9.49	41.57	102.99	0.332	Plasma Membrane
XM_014761623.3	*GmUBiA16*	402	44,363.28	9.94	36.09	109.93	0.496	Plasma Membrane
XM_041005679.1	*GmUBiA17*	412	46,579.91	9.55	54.66	106.67	0.397	Plasma Membrane
NM_001366991.1	*GmUBiA18*	402	45,250.3	9.61	48.84	105.97	0.446	Plasma Membrane
NM_001348662.1	*GmUBiA19*	392	43,502.4	9.3	35.8	114.67	0.589	Plasma Membrane
XM_003532557.5	*GmUBiA20*	398	44,124.58	8.73	35	101.91	0.417	Plasma Membrane
XM_003543880.5	*GmUBiA21*	397	44,097.67	8.98	36.77	102.64	0.369	Plasma Membrane
XM_041006158.1	*GmUBiA22*	230	25,921.12	9.84	36.66	124.96	0.861	Plasma Membrane
XM_026126145.2	*GmUBiA23*	277	30,997.81	9	35.8	110.47	0.627	Plasma Membrane
XM_041006200.1	*GmUBiA24*	293	32,271.04	8.8	41.52	103.45	0.566	Plasma Membrane
XM_020738038.1	*PeUBiA1*	403	44,033.25	9.13	50.67	105.56	0.309	Plasma Membrane
XM_020718723.1	*PeUBiA2*	475	51,887.23	9.19	46.28	93.75	0.315	Plasma Membrane
XM_020732720.1	*PeUBiA3*	471	51,393.55	8.93	50.25	93.27	0.293	Plasma Membrane
XM_020722467.1	*PeUBiA4*	374	40,536.16	8.79	31.78	109.09	0.305	Plasma Membrane
XM_020722914.1	*PeUBiA5*	318	35,194.82	9.15	29.75	128.87	0.874	Plasma Membrane
XM_020729489.1	*PeUBiA6*	331	37,301.06	9.47	38.54	109.94	0.569	Plasma Membrane
XM_020726705.1	*PeUBiA7*	388	42,121.47	8.98	36.94	96.29	0.465	Plasma Membrane
XM_020738474.1	*PeUBiA8*	257	29,003.29	9.88	39.1	102.88	0.594	Plasma Membrane
WP_010874074.1	*SsUBiA1*	292	31,645.74	6.54	34.17	130.27	0.899	Plasma Membrane
WP_010872404.1	*SsUBiA2*	308	34,410.07	9.02	28.08	134.87	0.853	Plasma Membrane
WP_010873507.1	*SsUBiA3*	324	35,208.44	6.05	19.93	116.23	0.546	Plasma Membrane
WP_010872655.1	*SsUBiA4*	307	33,246.9	7.08	29.88	116.91	0.639	Plasma Membrane
WP_010872804.1	*SsUBiA5*	316	34,857.68	9.13	27.37	128.73	0.703	Plasma Membrane
XM_043068307.1	*CrUBiA1*	379	40,389.72	8.97	37.32	98.05	0.186	Plasma Membrane
XM_001695355.2	*CrUBiA2*	443	45,902.39	7.7	54.22	87.43	0.197	Plasma Membrane
XM_001701536.2	*CrUBiA3*	383	41,448.11	8.49	28.62	103.26	0.298	Plasma Membrane
XM_043066342.1	*CrUBiA4*	322	32,758.75	9.01	24.07	127.39	0.987	Plasma Membrane
XM_001695289.2	*CrUBiA5*	370	39,347.15	9.37	41.36	111.41	0.595	Plasma Membrane
XM_043061866.1	*CrUBiA6*	445	43,763.78	9.17	41.98	100.54	0.659	Plasma Membrane
NM_001341624.1	*AtUBiA1*	407	44,603.53	9.23	35.01	96.34	0.137	Plasma Membrane
NM_130015.4	*AtUBiA2*	431	46,475.03	9.87	45.51	90.93	0.27	Plasma Membrane
NM_179653.4	*AtUBiA3*	393	43,908.96	9.74	41.9	113.36	0.56	Plasma Membrane
NM_115041.4	*AtUBiA4*	387	41,881.44	8.52	27.04	104.81	0.267	Plasma Membrane
NM_001084669.2	*AtUBiA5*	386	42,840.52	10.15	30.82	115.75	0.542	Plasma Membrane
NM_104743.3	*AtUBiA6*	287	31,054.75	9.18	30.63	123.66	0.706	Plasma Membrane
XM_023879341.2	*LsUBiA1*	407	45,328.19	9.19	42.95	94.91	0.008	Plasma Membrane
XM_023874978.2	*LsUBiA2*	439	47,603.05	9.46	34.16	92.51	0.171	Plasma Membrane
XM_023904443.2	*LsUBiA3*	379	41,111.8	8.73	29.3	102.51	0.267	Plasma Membrane
XM_023909767.2	*LsUBiA4*	395	44,741.91	9.62	44.83	107.85	0.443	Plasma Membrane
XM_023901890.2	*LsUBiA5*	395	44,761.73	9.58	47.37	111.77	0.637	Plasma Membrane
XM_023909580.2	*LsUBiA6*	399	45,145.59	9.46	42.71	102.68	0.418	Plasma Membrane
XM_023913555.2	*LsUBiA7*	334	37,050.93	9.57	33	123.17	0.505	Plasma Membrane
XM_023901886.2	*LsUBiA8*	416	46,609.98	9.66	34.11	117.88	0.42	Plasma Membrane
XM_023901043.2	*LsUBiA9*	384	42,777.3	9.67	34.2	107.71	0.52	Plasma Membrane
XM_023893904.2	*LsUBiA10*	409	45,553.65	9.65	33.88	116.31	0.36	Plasma Membrane
XM_042902057.1	*LsUBiA11*	479	53,606.59	8.9	41.44	110.33	0.252	Plasma Membrane
XM_023908970.2	*LsUBiA12*	300	33,379.45	9.38	30.33	118.93	0.467	Plasma Membrane
XM_023909766.2	*LsUBiA13*	398	43,239.93	9.37	32.4	111.73	0.442	Plasma Membrane
JAKOEK|L6452_mrna25818	*AlUBiA1*	261	28,191.91	9.11	31.89	107.24	0.448	Plasma Membrane
JAKOEK|L6452_mrna20277	*AlUBiA2*	497	53,962.3	9.68	40.86	90.78	0.059	Plasma Membrane
JAKOEK|L6452_mrna39700	*AlUBiA3*	395	43,986.85	9.64	45.87	115.47	0.631	Plasma Membrane
JAKOEK|L6452_mrna22236	*AlUBiA4*	433	48,739.19	9.51	35.29	110.81	0.37	Plasma Membrane
JAKOEK|L6452_mrna22240	*AlUBiA5*	406	45,958.41	9.29	40.73	108.97	0.55	Plasma Membrane
JAKOEK|L6452_mrna24308	*AlUBiA6*	384	42,580.01	9.69	37.47	107.94	0.563	Plasma Membrane
JAKOEK|L6452_mrna22828	*AlUBiA7*	405	43,960.54	9.58	36.37	112.2	0.404	Plasma Membrane
JAKOEK|L6452_mrna24308	*AlUBiA8*	384	42,580.01	9.69	37.47	107.94	0.563	Plasma Membrane
XM_002978823.2	*SmUBiA1*	373	40,376.75	9.6	48.87	102.01	0.186	Plasma Membrane
XM_002988434.2	*SmUBiA2*	373	40,295.69	9.42	43.85	102.28	0.245	Plasma Membrane
XM_024669487.1	*SmUBiA3*	427	45,928.2	9.12	35.26	89.77	0.181	Plasma Membrane
XM_024687212.1	*SmUBiA4*	427	45,908.12	9.36	34.92	87.73	0.134	Plasma Membrane
XM_024689909.1	*SmUBiA5*	390	42,593.43	8.71	29.5	101.87	0.26	Plasma Membrane
XM_024659843.1	*SmUBiA6*	390	42,686.6	8.71	28.82	102.62	0.257	Plasma Membrane
XM_002985500.3	*SmUBiA7*	388	42,192.01	9.75	54.4	114.48	0.678	Plasma Membrane
XM_002979186.2	*SmUBiA8*	363	40,060.19	9.59	27.46	117.66	0.552	Plasma Membrane
XM_024664099.1	*SmUBiA9*	363	40,066.16	9.59	26.63	117.11	0.546	Plasma Membrane
XM_024659418.1	*SmUBiA10*	353	39,419.15	9.89	58.54	97.88	0.271	Plasma Membrane
XM_024664337.1	*SmUBiA11*	325	34,633.71	8.48	24.56	120.09	0.632	Plasma Membrane
XM_024688846.1	*SmUBiA12*	269	28,648.87	7.6	22.61	120.45	0.778	Plasma Membrane
XM_002961810.2	*SmUBiA13*	366	39,183.08	8.93	40.02	114.92	0.546	Plasma Membrane
XM_002980750.2	*SmUBiA14*	366	39,231.12	8.93	40.02	114.13	0.542	Plasma Membrane
XM_024659390.1	*SmUBiA15*	346	37,322.91	9.36	48.66	109.91	0.518	Plasma Membrane
XM_024660820.1	*SmUBiA16*	346	37,496.13	9.37	47.18	109.62	0.527	Plasma Membrane
XM_015794109.2	*OsUBiA1*	417	45,579.06	9.49	46.47	96.47	0.14	Plasma Membrane
XM_015793105.2	*OsUBiA2*	418	45,307.63	9.03	58.59	94.4	0.143	Plasma Membrane
XM_015766928.2	*OsUBiA3*	435	46,539.75	9.61	44.01	90.6	0.175	Plasma Membrane
XM_015789024.2	*OsUBiA4*	404	44,501.31	9.85	52.66	108.94	0.51	Plasma Membrane
XM_015791419.2	*OsUBiA5*	379	41,421.94	9.99	40.71	111.56	0.565	Plasma Membrane
XM_015782164.2	*OsUBiA6*	376	40,578.99	8.23	29.2	104.63	0.289	Plasma Membrane
XM_015786513.2	*OsUBiA7*	404	44,899.63	9.15	42.67	102.92	0.378	Plasma Membrane
XM_015772366.2	*OsUBiA8*	369	39,316.01	9.57	37.68	104.99	0.521	Plasma Membrane
XM_026025964.1	*OsUBiA9*	205	22,086.31	6.64	39.67	99.51	0.26	Plasma Membrane
CYCAS_007186	*CpUBiA1*	317	34,783.72	8.74	36.19	108.36	0.332	Plasma Membrane
CYCAS_028733	*CpUBiA2*	478	51,943.14	9.31	41.34	91.72	0.13	Plasma Membrane
CYCAS_026755	*CpUBiA3*	427	47,516.28	9.81	42.87	110.77	0.51	Plasma Membrane
CYCAS_009776	*CpUBiA4*	399	44,110.11	10.01	33.16	114.46	0.637	Plasma Membrane
CYCAS_028820	*CpUBiA5*	451	49,065.88	7.62	36.65	110.29	0.264	Plasma Membrane
CYCAS_029627	*CpUBiA6*	402	43,142.54	9.39	38.72	106.02	0.43	Plasma Membrane
XM_035977189.1	*HaUBiA1*	400	44,671	9.5	36.43	103.15	0.455	Plasma Membrane
XM_022125636.2	*HaUBiA2*	403	44,765.53	9.31	39.62	96.38	0.389	Plasma Membrane
XM_022115079.2	*HaUBiA3*	374	40,492.14	8.87	33.66	108.05	0.294	Plasma Membrane
XM_035978370.1	*HaUBiA4*	458	50,808.91	8.07	36.01	122.42	0.572	Plasma Membrane
XM_022133493.2	*HaUBiA5*	398	43,348.1	9.43	36.1	109.05	0.456	Plasma Membrane
XM_022170044.2	*HaUBiA6*	446	48,452.02	9.84	36.17	90.81	0.13	Plasma Membrane
XM_022137525.2	*HaUBiA7*	388	42,826.65	9.46	35.46	101.11	0.211	Plasma Membrane
XM_022133910.2	*HaUBiA8*	397	44,132.01	9.64	45.68	113.93	0.642	Plasma Membrane
XM_022148176.2	*HaUBiA9*	372	41,976.64	9.5	39.1	114.52	0.39	Plasma Membrane
XM_022144901.2	*HaUBiA10*	375	41,704.12	9.81	33.01	114.21	0.601	Plasma Membrane
XM_024545301.1	*PpUBiA1*	526	58,058.76	9.09	35.24	95.48	−0.025	Plasma Membrane
XM_024539677.1	*PpUBiA2*	495	52,852.45	9.91	33.75	90.3	0.206	Plasma Membrane
XM_024546331.1	*PpUBiA3*	427	47,112.91	9.52	40.99	98.45	0.319	Plasma Membrane
XM_024523725.1	*PpUBiA4*	429	46,955.35	9.85	35.81	109.6	0.484	Plasma Membrane
XM_024533894.1	*PpUBiA5*	415	45,263.45	9.94	41.24	111.93	0.537	Plasma Membrane
XM_024536594.1	*PpUBiA6*	596	64,246.01	9.36	39.27	94.85	0.057	Plasma Membrane
XM_024541651.1	*PpUBiA7*	415	44,777.38	9.73	34.17	108.92	0.471	Plasma Membrane
XM_024541769.1	*PpUBiA8*	425	45,232.4	9.31	35.04	112.92	0.502	Plasma Membrane

## Data Availability

All the other data sets supporting the conclusions of this article are included within the article and its Appendix A.

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
