# Peer review of "Genome-Wide Identification and Expression Analysis of UBiA Family Genes Associated with Abiotic Stress in Sunflowers (Helianthus annuus L.)"

_ijms, 2023, doi:10.3390/ijms24031883_

Round 1

Reviewer 1 Report

This manuscript entitled " Genome-wide identification and expression analysis of UBiA family genes associated with abiotic stress in sunflower (Helianthus annuus L.)" could be good for publication in International Journal of Molecular Sciences.

This may be interesting, but some important points need to be resolved. Importantly, a study must provide a critical analysis of the data. In other words, you must assess whether specific data published really stand up to scientific scrutiny. In order to achieve the above, you must clearly define your specific aims and objectives. So in your study you must develop a critical appraisal of the state of the art. This is an essential element of any article. There are important scientific questions (both conceptual and methodological) which need to be addressed with the primary studies. A study must highlight this. The introduction, which is written in clear language, covers a number of relevant issues. Information are noteworthy, and not are correct supported by similar results from the specialty (see WOS: 000327818000032, WOS: 000353029900029, WOS: 000339050700030, WOS: 000229981900029).  Try to rewrite the abstract and conclusions, I also recommend the nuance of the introduction, the way of working is not very well explained, the procedure is tedious and unsustainable. For this reason, I recommend that the authors try to use more sustainable methodologies, the interpretation of the results can be improved/ reformulated,

Author Response

Response to Reviewer 1 Comments

Point 1: This manuscript entitled " Genome-wide identification and expression analysis of UBiA family genes associated with abiotic stress in sunflower (Helianthus annuus L.)" could be good for publication in International Journal of Molecular Sciences.

This may be interesting, but some important points need to be resolved. Importantly, a study must provide a critical analysis of the data. In other words, you must assess whether specific data published really stand up to scientific scrutiny. In order to achieve the above, you must clearly define your specific aims and objectives. So in your study you must develop a critical appraisal of the state of the art. This is an essential element of any article. There are important scientific questions (both conceptual and methodological) which need to be addressed with the primary studies. A study must highlight this. The introduction, which is written in clear language, covers a number of relevant issues. Information are noteworthy, and not are correct supported by similar results from the specialty (see WOS: 000327818000032, WOS: 000353029900029, WOS: 000339050700030, WOS: 000229981900029).  Try to rewrite the abstract and conclusions, I also recommend the nuance of the introduction, the way of working is not very well explained, the procedure is tedious and unsustainable. For this reason, I recommend that the authors try to use more sustainable methodologies, the interpretation of the results can be improved/ reformulated,

Response 1: Thanks for your suggestion. The rewriting of the abstract and conclusions has been completed and the results have been more concisely described and rearranged.

Reviewer 2 Report

line 41: ..members..

line 376: ..study..

line 390: ..vitamin E.. ?

line 434: Italics  ?

line 544, 555: the reference is not mentioned in the text...

Reviewer 3 Report

The manuscript "Genome-wide identification and expression analysis of UBiA family genes associated with abiotic stress in sunflower (Helianthus annuus L.) " described the conserved protein motifs and domains of UBiA genes in sunflower, and the expression pattern of the genes in various tissue under normal, plant stress and hormone stress conditions. These results provide a reference for future functional studies of the UBiA family genes. Most results make sense to me except the phylogenetic analysis of the UBiA genes. The authors constructed the phylogenetic trees using the neighbor-joining method and tree nodes were evaluated through 1000 repeated bootstrap analyses. But in Figures 2 and 3, there are no bootstrap values that are used to assess the reliability of sequence-based phylogeny for each branch. 

Please see below for other minor suggestions or questions.  

Line 199. Domain prediction is in Figure S3. The same question is in Line 203. The motif prediction should be in Figure S2. 

Line 285. Please indicate here and in the legend of Figure 6 that “14-day-old seedlings” were used to test the response of hormone signals. 

Line 329. There is no experimental evidence of protein interaction of UBiA1 and UBiA5, especially between UBiA5 and the NAC transcription factor. So, the result might or might not be true.  

Line 389. Literature to support this. 

Round 2

Reviewer 1 Report

This manuscript entitled " Genome-wide identification and expression analysis of UBiA family genes associated with abiotic stress in sunflower (Helianthus annuus L.)"; could be good for publication in International Journal of Molecular Sciences.

Author Response

Thank you for your approval. Your comments and suggestions had helped us a lot to improve this article.

Reviewer 3 Report

Please add bootstrap values onto your phylogenic tree in Figures 2, 3 and other figures if applicable. Please move Figure 8 from the main text because the interactions are not your main result. Readers might be confused about how an isopentenyltransferase interacts with a transcription factor, and the prediction based on STRING might not provide strong evidence to support your point.

Author Response

Response to Reviewer 3 Comments

Point 1: Please add bootstrap values onto your phylogenic tree in Figures 2, 3 and other figures if applicable. Please move Figure 8 from the main text because the interactions are not your main result. Readers might be confused about how an isopentenyltransferase interacts with a transcription factor, and the prediction based on STRING might not provide strong evidence to support your point.

Response 1: Thank you for your suggestion. We added bootstrap values onto phylogenic tree in Figures 2, 3. We had moved Figure 8 to the supplementary material from the main text.